

# Murine pluripotent stem cells that escape differentiation inside teratomas maintain pluripotency

Yangli Pei[1,2,*], Liang Yue[2,*], Wei Zhang[2], Jinzhu Xiang[2], Zhu Ma[3] and Jianyong Han[2]

[1] State Key Laboratory of Animal Nutrition, Institute of Animal Sciences, Chinese Academy of Agricultural Sciences, Beijing, China
[2] State Key Laboratories for Agrobiotechnology, College of Biological Sciences, China Agricultural University, Beijing, China
[3] Beijing Dairy Cattle Center, Beijing, China
* These authors contributed equally to this work.

## ABSTRACT

**Background**. Pluripotent stem cells (PSCs) offer immense potential as a source for regenerative therapies. The teratoma assay is widely used in the field of stem cells and regenerative medicine, but the cell composition of teratoma is still elusive.

**Methods**. We utilized PSCs expressing enhanced green fluorescent protein (EGFP) under the control of the *Pou5f1* promoter to study the persistence of potential pluripotent cells during teratoma formation *in vivo*. OCT4-MES (mouse embryonic stem cells) were isolated from the blastocysts of 3.5-day OCT4-EGFP mice (transgenic mice express EGFP cDNA under the control of the *Pou5f1* promoter) embryos, and TG iPS 1-7 (induced pluripotent stem cells) were generated from mouse embryonic fibroblasts (MEFs) from 13.5-day OCT4-EGFP mice embryos by infecting them with a virus carrying OCT4, SOX2, KLF4 and c-MYC. These pluripotent cells were characterized according to their morphology and expression of pluripotency markers. Their differentiation ability was studied with *in vivo* teratoma formation assays. Further differences between pluripotent cells were examined by real-time quantitative PCR (qPCR).

**Results**. The results showed that several OCT4-expressing PSCs escaped differentiation inside of teratomas, and these escaped cells (MES-FT, GFP-positive cells separated from OCT4-MES-derived teratomas; and iPS-FT, GFP-positive cells obtained from teratomas formed by TG iPS 1-7) retained their pluripotency. Interestingly, a small number of GFP-positive cells in teratomas formed by MES-FT and iPS-FT (MES-ST, GFP-positive cells isolated from MES-FT-derived teratomas; iPS-ST, GFP-positive cells obtained from teratomas formed by iPS-FT) were still pluripotent, as shown by alkaline phosphatase (AP) staining, immunofluorescent staining and PCR. MES-FT, iPS-FT, MES-ST and iPS-ST cells also expressed several markers associated with germ cell formation, such as *Dazl*, *Stella* and S*tra8*.

**Conclusions**. In summary, a small number of PSCs escaped differentiation inside of teratomas, and these cells maintained pluripotency and partially developed towards germ cells. Both escaped PSCs and germ cells present a risk of tumor formation. Therefore, medical workers must be careful in preventing tumor formation when stem cells are used to treat specific diseases.

Corresponding author
Jianyong Han, hanjy@cau.edu.cn

## INTRODUCTION

Pluripotent stem cells (PSCs), including embryonic stem cells (ESCs) and induced pluripotent stem cells (iPSCs), have the potential to differentiate into all cell types of the body *in vitro* through embryoid body formation or *in vivo* through teratoma formation. Due to these characteristics, stem cells provide an option for treating a multitude of clinical problems, such as myocardium damage after heart infarction, spinal cord damage after mechanical injury, brain damage after stroke, age-related macular degeneration of the retina, liver damage, extensive skin burns, Parkinson's disease, and diabetes (*Abdelalim et al., 2014*; *Lodi, Iannitti & Palmieri, 2011*; *Orlic et al., 2001*; *Ratajczak, Bujko & Wojakowski, 2016*).

When transplanted into immune-compromised mice, undifferentiated PSCs can form teratomas, consisting of multiple tissue types derived from all three germ layers (*Przyborski, 2005*; *Takahashi & Yamanaka, 2006*). As such, there have been many efforts to differentiate pluripotent cells to cells with medical applications in an *in vivo* developmental environment. For example, neural stem cells (NSCs) have been differentiated *in vivo* through teratoma formation, and pure NSC populations exhibit properties similar to those of brain-derived NSCs (*Hong et al., 2016*). Similarly, fully functional and engraftable hematopoietic stem/progenitor cells (HSPCs), along with functional myeloid and lymphoid cells, have been isolated from teratomas when human iPSCs were transplanted into immunodeficient mice (*Amabile et al., 2013*; *Suzuki et al., 2013*). In addition, the teratoma assay can be applied to assess the safety of human PSC-derived cell populations that are used for therapeutic application since a small number of undifferentiated cells contaminating a given transplant material can be efficiently detected by their multi-lineage differentiation ability (*Stachelscheid et al., 2013*).

However, the intrinsic self-renewal and pluripotency qualities of PSCs that make them therapeutically promising are responsible for an equally fundamental tumorigenic risk (*Lee et al., 2013*). Studies on teratomas will contribute to a better understanding of their stepwise development processes and underlying molecular mechanisms and may provide helpful information for the development of tissue engineering technologies (*Aleckovic & Simon, 2008*). These facts prompted us to address the additional characteristics of teratoma growth and differentiation after PSCs injection.

In the present study, we aimed to isolate OCT4-expressing cells that escaped differentiation inside of growing teratomas and to determine whether OCT4-expressing cells still possess self-renewal and pluripotency abilities.

## MATERIALS & METHODS

All animal experiments were approved by the Animal Care and Use Committees of the State Key Laboratories for Agrobiotechnology, College of Biological Sciences, China Agricultural

University (Approval number: SKLAB-2016-05-01). Briefly, mice were bred in a 12/12 h light/dark period and sacrificed by cervical vertebra dislocation.

## Mouse strains

OCT4-GFP transgenic mice (Model Animal Research Center of Nanjing University) express EGFP (enhanced green fluorescence protein) cDNA under the control of the *Pou5f1* promoter, which is active in pluripotent stem cells. This strain is useful for isolating pluripotent stem cells, as they specifically express green fluorescent protein. These OCT4-GFP transgenic mice were the source of the OCT4-MES and OG2 MEFs (mouse embryonic fibroblasts of 13.5-day OCT4-EGFP mice embryos) used in this study.

## Derivation of MES and generation of iPSCs

To obtain OCT4-MES, uteri containing E3.5 embryos were isolated from timed pregnancies and transferred individually to the wells of a 24-well plate with irradiated mouse embryonic fibroblast (MEF) feeders. After five days of incubation, embryo outgrowths were separated from trophectoderm, individually picked, and expanded in MES medium (Dulbecco's modified eagle medium (DMEM) supplemented with 15% fetal bovine serum (FBS), L-glutamine, nonessential amino acids, β-mercaptoethanol, and 1,000 U/ml leukemia inhibitory factor).

OG2 MEFs were cultured in MEF medium (DMEM supplemented with 10% FBS, L-glutamine and nonessential amino acids); infected with retroviruses generated from pMX retroviral vectors encoding mouse *Pou5f1*, *Sox2*, *Klf4* and *c-Myc*; and cultured on irradiated MEF feeder cells in MES medium. Subsequently, a single ESC-like colony was individually picked and expanded on feeders to establish stable lines. Both OCT4-MES and iPSCs originated from male embryos. Additional details can be found in our previous study (*Pei et al., 2015*).

## Immunofluorescence

Cells were fixed with 4% paraformaldehyde, permeabilized with 0.1% Triton X-100, and blocked with 2% BSA. The cells were then stained with primary antibodies against OCT4 (ab19857, 1:500; Abcam, Cambridge, UK), SOX2 (ab97959, 1:1,000; Abcam, Cambridge, UK), NANOG (ab80892, 1:500; Abcam, Cambridge, UK) and SSEA1 (ab16285, 1:200; Abcam, Cambridge, UK), followed by staining with the respective secondary antibodies conjugated to Alexa Fluor (A-11008, A-11037, A-21044, 1:1,000; Invitrogen, Carlsbad, CA, USA). Finally, cells were counterstained with DAPI (D9542, Sigma, St. Louis, MO, USA).

## RNA purification and cDNA preparation

Feeders were removed by plating ESCs on a gelatin-coated dish for 30 min, and unattached cells were collected by centrifugation. Total RNA was extracted from pure PSCs using Trizol reagent according to the manufacturer's instructions (Invitrogen, Carlsbad, CA, USA). RNA was reverse-transcribed using oligo-dT and M-MLV Reverse Transcriptase (Promega, Madison, WI, USA).

**Table 1  Sequence of primers used in this study.**

| Gene | | Sequence (5′ − 3′) |
|---|---|---|
| Gapdh | Forward | AGGTCGGTGTGAACGGATTTG |
| | Reverse | TGTAGACCATGTAGTTGAGGTCA |
| β-tubulin | Forward | TGAGGCCTCCTCTCACAAGTA |
| | Reverse | CCGCACGACATCTAGGACTG |
| EF1α | Forward | GTGTTGTGAAAACCACCGCT |
| | Reverse | AGGAGCCCTTTCCCATCTCA |
| Pou5f1 | Forward | GTTGGAGAAGGTGGAACCAA |
| | Reverse | CTCCTTCTGCAGGGCTTTC |
| Sox2 | Forward | AAGGGTTCTTGCTGGGTTTT |
| | Reverse | AGACCACGAAAACGGTCTTG |
| Utf1 | Forward | GTCCGGACCCTTCGATAACC |
| | Reverse | CTCGGCCTCTTGCTCCAC |
| Nanog | Forward | TTCTTGCTTACAAGGGTCTGC |
| | Reverse | AGAGGAAGGGCGAGGAGA |
| Rex1 | Forward | CAGTTCGTCCATCTAAAAAGGGAGG |
| | Reverse | TCTTAGCTGCTTCCTTGAACAATGCC |
| Tbx3 | Forward | ATCGCCGTTACTGCCTATCA |
| | Reverse | TGCAGTGTGAGCTGCTTTCT |
| Lin28a | Forward | GTCTTTGTGCACCAGAGCAAG |
| | Reverse | ATGGATTCCAGACCCTTGGC |
| Nr5a2 | Forward | TAGGACCGGAAAGCGTCTGC |
| | Reverse | GCTTCCGTCTCCACTTTGGG |
| Dazl | Forward | GCCCGCAAAAGAAGTCTGTG |
| | Reverse | ACCAACAACCCCCTGAGATG |
| Stella | Forward | GAGAAGACTTGTTCGGATTGAGC |
| | Reverse | CATCGTCGACAGCCAGGG |
| Stra8 | Forward | CTCCTCCTCCACTCTGTTGC |
| | Reverse | GCGGCAGAGACAATAGGAAG |
| Vasa | Forward | ACCAAGATCAGGGGACACAG |
| | Reverse | TAACCACCTCGACCACTTCC |

## Real-time quantitative PCR

qPCR was performed on a LightCycler 480 II Real-Time PCR System (Roche, Basel, Switzerland) using the LightCycler 480 SYBR Green I Master Mix (Roche, Basel, Switzerland 4887352001). The qPCR data was analyzed using the comparative CT ($2^{-\Delta\Delta CT}$) method as the description by *Livak & Schmittgen (2001)*. The ΔCT was calculated using Gapdh, EF1α and β-tubulin as internal control.

The primers used for qPCR and PCR are listed in Table 1.

## Teratoma production and analysis

Approximately $1\times10^6$ PSCs were suspended in 150 μl of PBS (phosphate buffered solution) and subcutaneously injected into the hind limb of NOD/SCID mice to form teratomas. Three weeks after injection, the teratomas were harvested, fixed overnight with

4% paraformaldehyde, embedded in paraffin, sectioned, HE stained or immunostained (primary antibodies against GFP, Cat. 2956, 1: 200 (Cell Signaling Technology, Danvers, MA, USA); Biotin-Streptavidin horseradish peroxidase detection kit, Cat. SP-9001 (Beijing Zhongshan Golden Bridge Biotechnology Company, Beijing, China)), and analyzed.

### Statistical analysis

All results are presented as the mean ± standard deviation. Results were statistically analyzed using SAS (Statistics Analysis System) program. Significance of differences between samples was determined (at the significance level $p < 0.05$) using Kruskal–Wallis test.

## RESULTS

### Both OCT4-MES and TG iPS 1-7 are pluripotent

OCT4-EGFP mice express green fluorescent protein under the control of the pluripotency-associated *Pou5f1* promoter and are widely used to study the function of PSCs (*Pei et al., 2015*). These mice were used to generate mouse embryonic stem cells (MES) and iPSCs. OCT4-MES were isolated from the blastocysts of 3.5-day OCT4-EGFP mice embryos, while other mice were selected to prepare MEFs after day 13.5. The isolated MEFs were used to generate iPSCs by infecting them with a virus carrying OCT4, SOX2, KLF4 and c-MYC. Then, TG iPS 1-7 was selected from the isolated iPSC clones.

 Both OCT4-MES and TG iPS 1-7 were maintained on feeder cells in the presence of leukemia inhibitory factor. They both exhibited typical MES-like morphologies (Figs. 1A and 1B). Immunofluorescent staining confirmed the expression of the three master transcription factors (OCT4, NANOG and SOX2) as well as ESC-specific surface marker SSEA-1 in OCT4-MES and TG iPS 1-7 (Figs. 1C–1J). The PCR results further demonstrated that these cells expressed pluripotency marker genes, including *Pou5f1*, *Sox2*, *Nanog*, *Rex1*, *Tbx3*, *Nr5a2, Utf1* and *Lin28a* (Fig. 2). Next, *in vivo* teratoma formation assays were performed to further validate the pluripotency of OCT4-MES and TG iPS 1-7. Approximately $1 \times 10^6$ PSCs were suspended in 150 μl of PBS and injected into non-obese diabetic/severe combined immunodeficient (NOD/SCID) mice to form teratomas. Three weeks after injection, OCT4-MES and TG iPS 1-7 formed teratomas *in vivo*, and hematoxylin and eosin (H&E) staining confirmed the formation of all three germ layers in each teratoma (Figs. 3A–3F). These results revealed that OCT4-MES and TG iPS 1-7 were pluripotent. Interestingly, we observed OCT4-positive cells growing in clusters in the teratoma masses formed by OCT4-MES and TG iPS 1-7 (Figs. 3G, 3H).

### OCT4-positive cells from OCT4-MES and TG iPS 1-7 teratomas have self-renewal and pluripotency qualities

To quantify the fraction of OCT4-positive pluripotent cells in teratomas generated by OCT4-MES and TG iPS 1-7, we cut the teratomas into pieces and digested them with trypsin and then cultured the cells in MEF medium. Three days later, we found that most of these cells separated from OCT4-MES and that TG iPS 1-7-derived teratomas had the morphology of mouse embryonic cells, but a small number of cells were round and expressed GFP (Figs. 4A, 4B, 4D and 4E). After picking these cells and culturing them

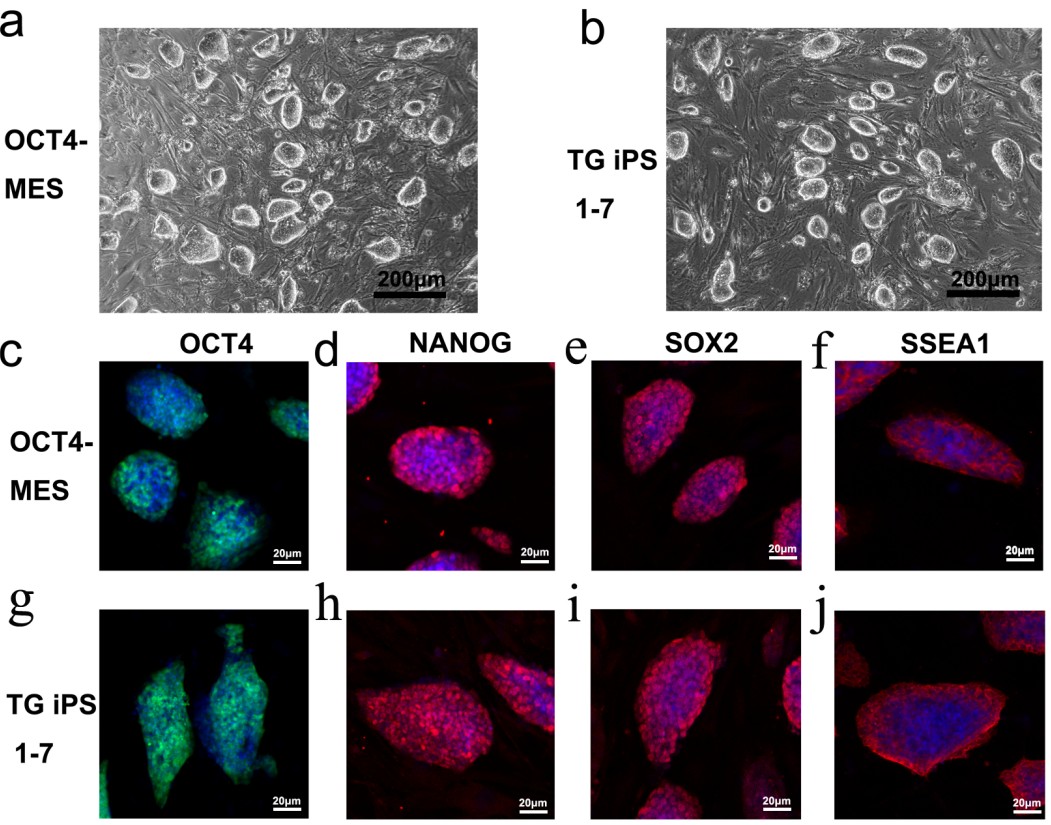

**Figure 1 OCT4-MES and TG iPS 1-7 are pluripotent.** (A and B) Phase contrast images of OCT4-MES (A) and TG iPS 1-7 (B). Both types of cells exhibit typical MES-like morphologies. (C–J) Immunofluorescent staining showing that both OCT4-MES (C–F) and TG iPS 1-7 (G–J) express the pluripotency markers OCT4 (C and G), NANOG (D and H), SOX2(E and I) and SSEA1 (F and J).

in MES medium, we found that they had typical MES-like morphologies, and they were AP-positive (Figs. 4C, 4F). We named these cells MES-FT and iPS-FT, which were derived from OCT4-MES and TG iPS 1-7, respectively. OCT4-expressing MES-FT and iPS-FT cells were grown in the presence of leukemia inhibitory factor, and they expressed pluripotency marker genes, including *Pou5f1*, *Sox2*, *Nanog*, *Rex1*, *Tbx3*, *Nr5a2*, *Utf1* and *Lin28a* (Fig. 2). The immunostaining results showed that these colonies were positive for OCT4, NANOG, SOX2 and SSEA-1 (Figs. 5A–5H). We performed *in vivo* teratoma formation assays to further validate the pluripotency of MES-FT and iPS-FT. MES-FT and iPS-FT formed teratomas *in vivo*, and the hematoxylin and eosin (H&E) staining results confirmed the formation of all three germ layers in each teratoma (Figs. 5I–5N). As in the results described above, there were also OCT4-positive pluripotent cells in the teratomas formed by MES-FT and iPS-FT (Figs. 5O, 5P).

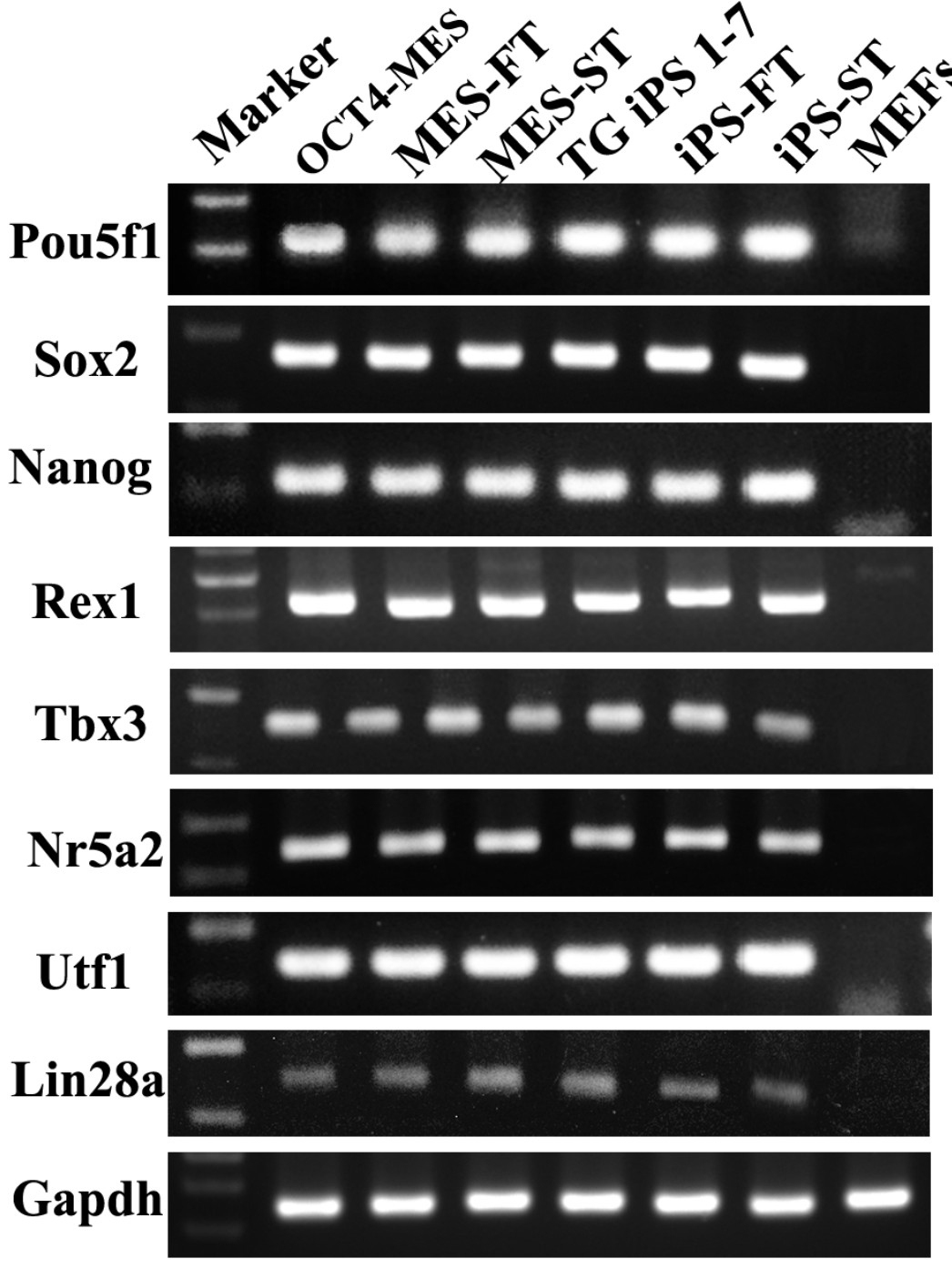

**Figure 2** **OCT4-MES, MES-FT, MES-ST, TG iPS 1-7, iPS-FT and iPS-ST express pluripotency genes.**
Expression of pluripotency marker genes was evaluated by PCR, showing that OCT4-MES, MES-FT, MES-ST, TG iPS 1-7, iPS-FT and iPS-ST all express pluripotency marker genes, including *Pou5f1*, *Sox2*, *Nanog*, *Rex1*, *Tbx3*, *Nr5a2*, *Utf1* and *Lin28a*.

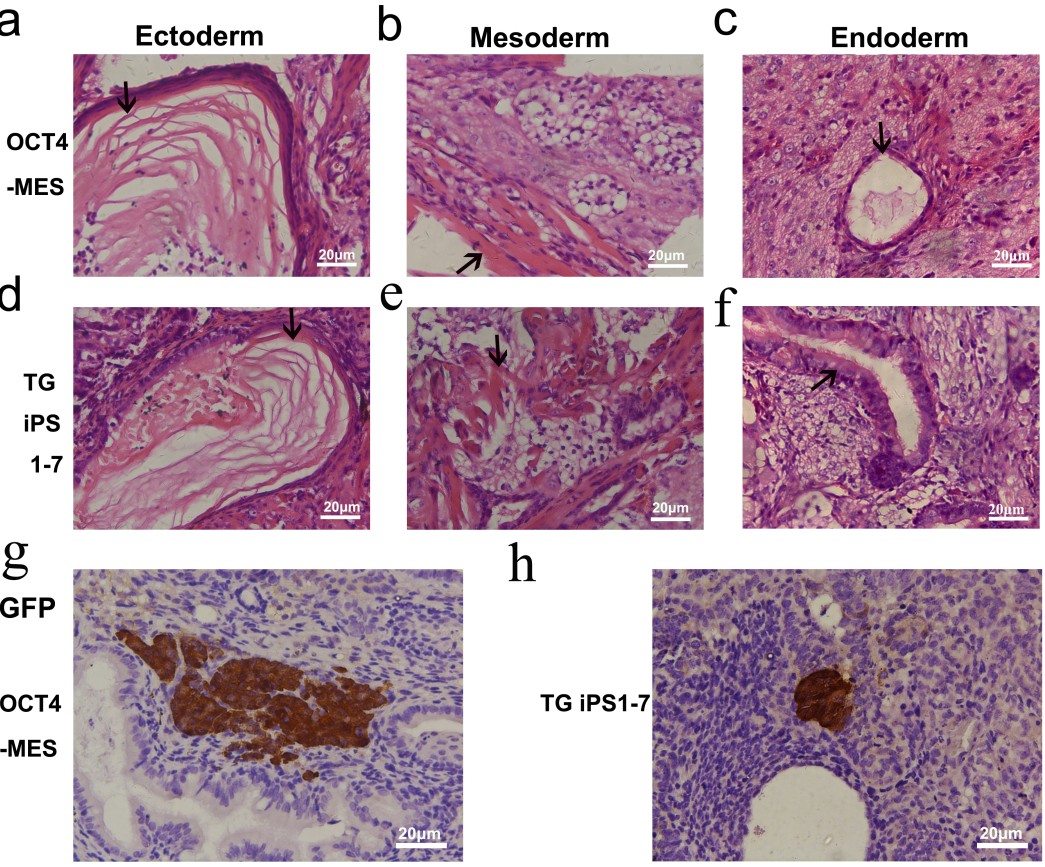

**Figure 3  GFP-positive pluripotent cells are present in teratomas generated by OCT4-MES and TG iPS 1-7.** (A–F) Hematoxylin and eosin staining of teratomas derived from OCT4-MES (A–C) and TG iPS 1-7 (D–F). Products of all three germ layers are seen in the image: Ectoderm: epidermis with keratin (A and D). Mesoderm: smooth muscle (A and E). Endoderm: gastrointestinal lining cells/glands (C and F). Specified cells are indicated by arrows. (G and H) Immunohistochemistry to detect the presence of GFP-positive pluripotent cells in teratomas generated by OCT4-MES (G) and TG iPS 1-7 (H). GFP-positive pluripotent cells were stained in brown with anti-GFP primary antibodies.

## OCT4-positive cells from MES-FT and iPS-FT teratomas are still pluripotent

We discovered several round and bright cells expressing OCT4-GFP under a microscope in cells separated from teratomas formed by MES-FT and iPS-FT cells (Figs. 6A, 6B, 6D and 6E). These round and bright cells formed AP positive clones (Figs. 6C, 6F). We named these cells MES-ST and iPS-ST. OCT4-expressing MES-ST and iPS-ST cells also expressed pluripotency marker genes, including *Pou5f1*, *Sox2*, *Nanog*, *Rex1*, *Tbx3*, *Nr5a2, Utf1* and *Lin28a* (Fig. 2), and the immunostaining results demonstrated that they expressed the stemness regulators OCT4, NANOG, SOX2 and SSEA-1 (Figs. 6G–6N).

The above results showed that OCT4-MES, TG iPS 1-7, MES-ST, iPS-ST, MES-ST and iPS-ST had pluripotency characteristics. However, MES-FT, iPS-FT, MES-ST and iPS-ST were survivors of the differentiation environment, so we wanted to know whether there were differences among these cells. Thus, we next investigated their differences.

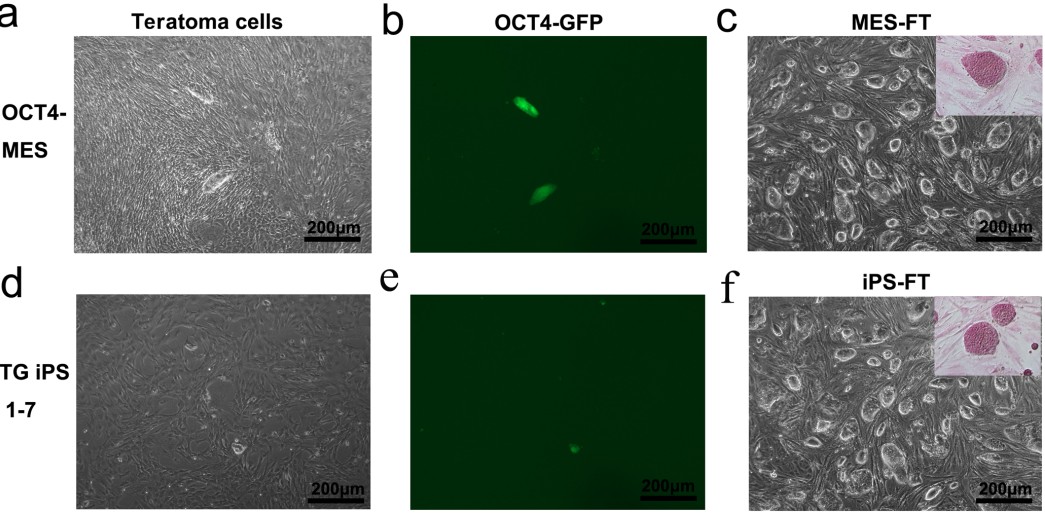

**Figure 4** **OCT4-positive pluripotent cells isolated from teratomas have typical mouse embryonic cell morphology.** (A, B, D and E) A small number of OCT4-GFP positive cells were found among teratoma cells generated by OCT4-MES (A and B) and TG iPS 1-7 (D and F) cultured in MEF medium. (C and F) MES-FT (C) and iPS-FT (F) have typical mouse embryonic cell morphology and are AP-positive when cultured in MES medium.

## OCT4-positive cells separated from teratomas express germ cell marker genes

To explore the gene expression patterns of OCT4-MES, TG iPS 1-7, MES-FT, iPS-FT, MES-ST and iPS-ST, cDNA was prepared from these cells without feeders for gene expression analysis. First, we detected the expression of pluripotency genes. When normalized to the values for OCT4-MES cells, the expression level of *Pou5f1* was higher in MES-FT, and that of *Lin28a* was higher in both MES-FT and MES-ST cells, but there were no differences in the *Nanog* expression levels between these three cell lines (Fig. 7A). When normalized to the values for TG iPS 1-7 cells, iPS-FT and iPS-ST both highly expressed *Pou5f1* and *Nanog* (Fig. 7B). However, there were no differences in the expression level of *Lin28a* (Fig. 7B). The expression of pluripotency marker genes in these cells varied slightly, but they were all within reasonable levels. Thus, these cell types were all pluripotent.

Previous results have shown that PSCs that escape from differentiation inside of embryonic bodies express several markers associated with germ cell formation (*Attia et al., 2014*). As such, we further assayed the differences between the expression levels of important germ cell-specific genes (*Dazl*, *Stella*, *Stra8*, *Vasa*) in MES-FT, iPS-FT, MES-ST and iPS-ST (Figs. 7C, 7D). When normalized to the values for OCT4-MES, *Dazl* and *Stella* were more highly expressed in MES-FT cells, and the expression level of S*tra8* was elevated nine-fold and ten-fold in MES-FT and MES-ST, respectively.

Similarly, iPS-FT and iPS-ST highly expressed *Dazl*, *Stra8* and *Vasa* than TG iPS 1-7. iPS-FT also highly expressed *Stella*. The above results show that OCT4-positive cells separated from teratomas have elevated expression of several markers associated with germ cell formation, such as *Dazl*, *Stella* and S*tra8*.

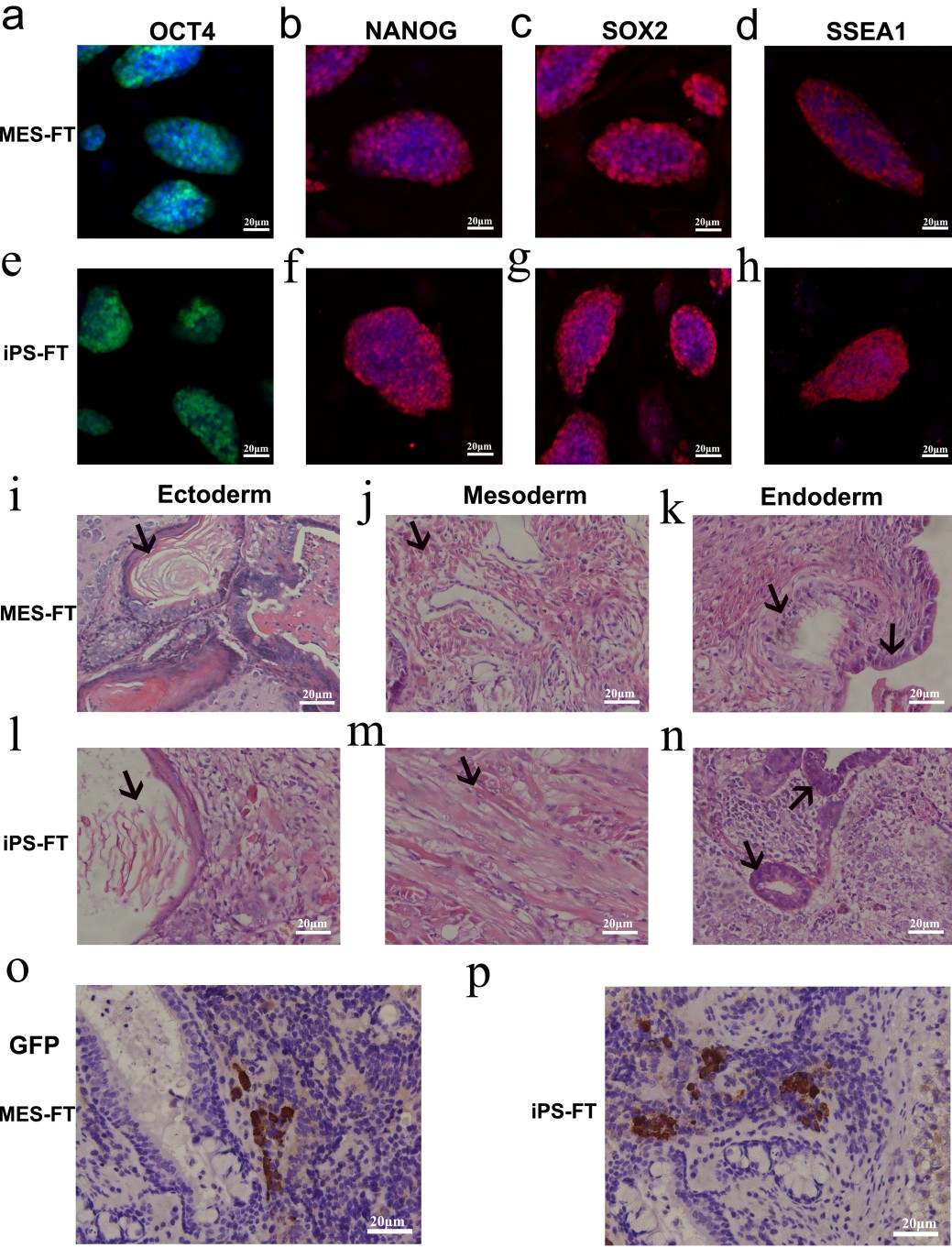

**Figure 5** **MES-FT and iPS-FT own self-renewal and pluripotency ability.** (A–H) Immunofluorescent staining of pluripotency markers OCT4, NANOG, SSEA1 and SOX2 in MES-FT (A–D) and iPS-FT (E–H). Both types of cells expressed all four markers. (I–N) Hematoxylin and eosin staining of teratomas derived from MES-FT (I–K) and iPS-FT (L–N). Products of all three germ layers are seen in the image: Ectoderm: epidermis with keratin (I and L). Mesoderm: smooth muscle (J and M). Endoderm: gastrointestinal lining cells/glands (K and N). Specified cells are indicated by arrows. (O and P) Immunohistochemistry to detect the presence of GFP-positive pluripotent cells in teratomas generated by MES-FT (O) and iPS-FT (P). GFP-positive pluripotent cells were stained in brown with anti-GFP primary antibodies.

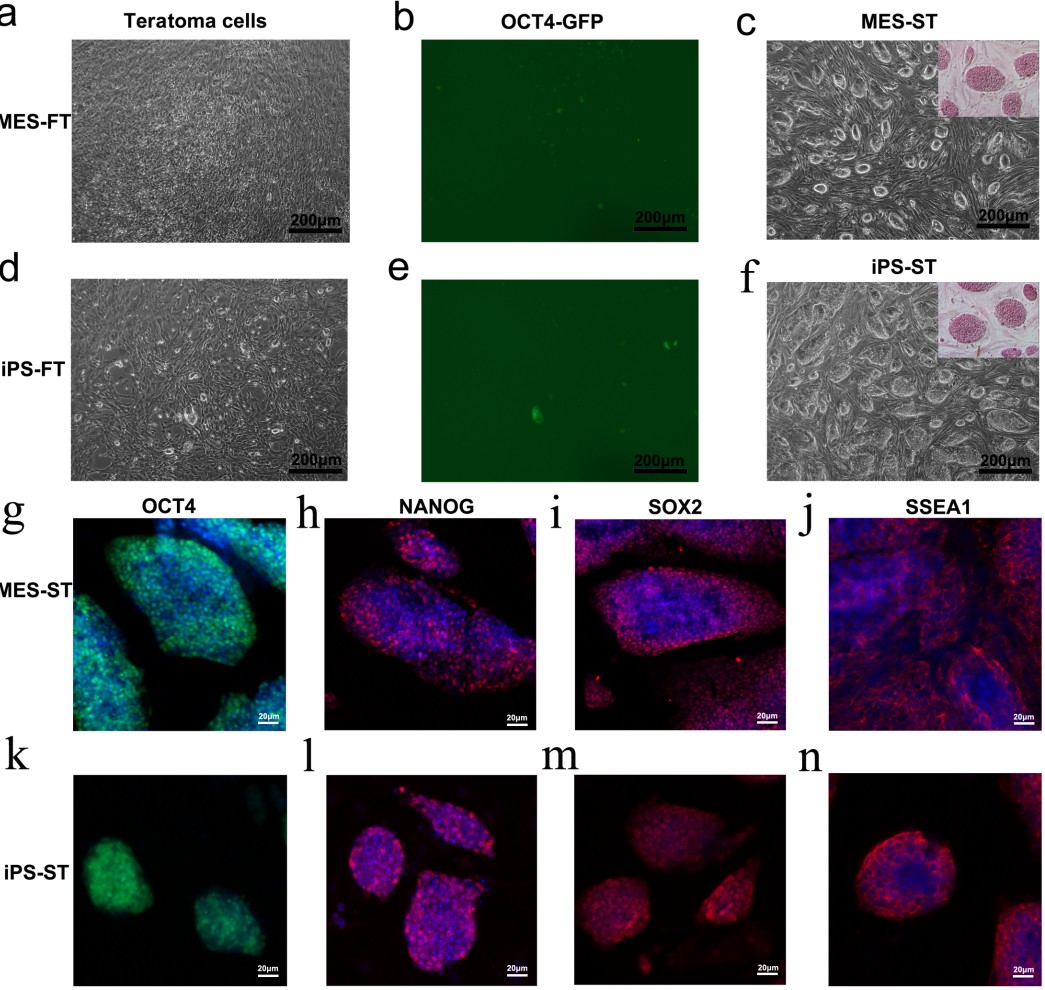

**Figure 6  MES-ST and iPS-ST owned PSCs characteristic.** (A, B, D and E) A small number GFP-positive cells were found among teratoma cells generated by MES-FT (A and B) and iPS-FT (D and E) cultured in MEF medium. (C and F) MES-ST (C) and iPS-ST (F) have typical mouse embryonic cell morphology and are AP-positive when cultured in MES medium. (G–K) Immunofluorescent staining of pluripotency markers OCT4, NANOG, SSEA1 and SOX2 in MES-ST (G–J) and iPS-ST (K–N). Both MES-ST (G–J) and iPS-ST (K–N) expressed all four markers.

## DISCUSSION

ESCs and iPSCs are characterized by their ability to develop into any cell type of the adult organism. As such, they can be widely applied to the treatment of many diseases. This is especially true for iPSCs, as they do not present ethical issues.

A previous report demonstrated the presence of undifferentiated human ESCs expressing the surface marker CD133 (*Ritner & Bernstein, 2010*). However, no additional research has been performed to investigate the characteristics of those undifferentiated cells in teratomas. Therefore, in this study, we isolated OCT4-GFP positive cells, MES-FT and iPS-FT, from teratomas generated by OCT4-MES and TG iPS 1-7, respectively. MES-FT and

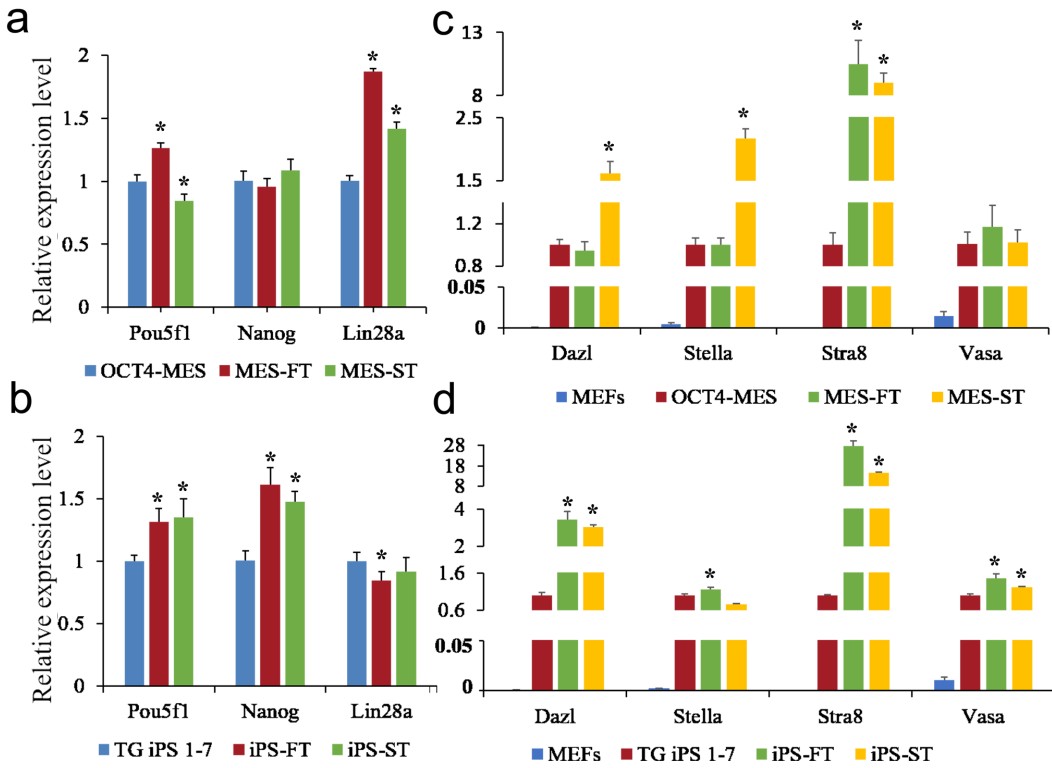

**Figure 7** **OCT4-MES, TG iPS 1-7, MES-FT, iPS-FT, MES-ST and iPS-ST express pluripotency genes; MES-FT, iPS-FT, MES-ST and iPS-ST more highly express several markers associated with germ cell formation.** (A) The expression levels of *Pou5f1*, *Nanog* and *Lin28a* in OCT4-MES, MES-FT and MES-ST were determined by qPCR. Both MES-FT and MES-ST highly expressed *Lin28a*, and MES-FT also highly expressed *Pou5f1*. (B) The expression levels of *pou5f1*, *Nanog* and *Lin28a* in TG iPS 1-7, iPS-FT and iPS-ST were determined by qPCR. iPS-FT and iPS-ST highly expressed *pou5f1* and *Lin28a*. (C) The expression levels of *Dazl*, *Stella*, *Stra8* and *Vasa* in Oct4-MES, MES-FT and MES-ST were determined by qPCR. Both MES-FT and MES-ST highly expressed *Stra8*. MES-ST also highly expressed *Dazl* and *Stella*. (D) The expression levels of *Dazl*, *Stella*, *Stra8* and *Vasa* in TG iPS 1-7, iPS-FT and iPS-ST were determined by qPCR. iPS-FT and iPS-ST highly expressed *Dazl*, *Stra8* and *Vasa*. iPS-FT also highly expressed *Stella*. Relative expression was quantified using the comparative threshold cycle (Ct) method ($2^{-\Delta\Delta Ct}$). $n = 3$, Gapdh, EF1α and β-tubulin were used as references. *, $p < 0.05$.

iPS-FT exhibit MES-like morphologies, express pluripotency marker genes and proteins, and can generate all three germ layers in an *in vivo* differentiation model. We discovered that there were still pluripotent cells in the teratomas formed by MES-FT and iPS-FT, so we separated them from the teratoma mass and named them MES-ST and iPS-ST. Further study confirmed that these isolated cells (MES-ST and iPS-ST) retained pluripotency and were capable of differentiation. From these results, it can be inferred that a subset of PSCs escape differentiation during *in vivo* differentiation, and the escaped cells retain their PSC characteristics in the appropriate environment. Since the escaped PSCs (MES-FT, iPS-FT, MES-ST and iPS-ST) still possessed PSC-like characteristics, these cells may progress to tumor formation at an undefined later time point.

*Bottai et al. (2010)* reported that they used $5 \times 10^5$ undifferentiated murine ESCs to cure spinal cord injury. However, some of the transplanted ESCs were found as dense aggregates in the tissue. This result supports our view that ESCs can be maintained *in vivo*. Another study showed that transplantation of $1 - 2 \times 10^6$ MES cells into SV129 mice led to tumor formation in 100% of cases, whereas transplantation of $5 \times 10^5$ cells produced tumors in two of six mice and transplantation of $1 \times 10^5$ ESCs gave rise to tumor formation in one of six transplanted mice within 100 days (*Dressel et al., 2008*). It can be deduced that there is likely a niche within teratomas that nurse PSCs, and the number of cells determines the niche environment. The more PSCs used for transplantation, the higher probability of tumor formation.

The escaped PSCs (MES-FT, iPS-FT, MES-ST and iPS-ST) showed slight similarities to primordial germ cells (PGCs), as shown by the high expression of *Pou5f1*, *Dazl*, *Stella* and *Stra8* in MES-FT, MES-ST, iPS-FT, and iPS-ST. *Pou5f1*, *Dazl*, *Stella*, *Stra8* and *Vasa* are well-known germ cell markers, and they are also commonly expressed in ESCs (*Cauffman et al., 2005*; *Kehler et al., 2004*; *Tedesco et al., 2009*; *Toyooka et al., 2000*; *Wongtrakoongate et al., 2013*). *Stra8* is required for the chromosomal program of meiotic prophase (*Soh et al., 2015*). *Dazl*, an intrinsic meiotic competence factor, is required for *Stra8*-mediated initiation of meiosis in germ cells (*Lin et al., 2008*). Overexpression of *Stra8* and *Dazl* genes promotes the transdifferentiation of mesenchymal stem cells and ESCs *in vitro* toward PGCs (*Li et al., 2017*; *Shi et al., 2014*). The elevated expression of *Pou5f1*, *Dazl*, *Stella* and *Stra8* might indicate that the GFP positive cells separated from teratomas partially develop towards germ cells. This suggests that it is possible to isolate PGCs from teratoma differentiation models.

## CONCLUSIONS

In summary, we found a small number of OCT4-expressing PSCs that escaped differentiation inside teratomas. The escaped cells kept their unique properties of self-renewal and pluripotency and were able to form teratomas *in vivo*. They also expressed several markers associated with germ cell formation, such as *Pou5f1*, *Dazl*, *Stella* and *Stra8*, suggesting that these cells may partially differentiate into germ cells. Therefore, this study serves as a warning that medical workers using stem cells to treat specific diseases must pay careful attention to prevent tumor formation because OCT4-expressing cells retain pluripotency, and it is feasible to isolate germ cells from teratomas. This study of PSCs that remain undifferentiated within teratomas has provided critical information for further investigation of the applications of stem cell therapy and for obtaining germ cells from *in vivo* differentiation models.

### Funding

This study was funded by grants from the National Key Research and Development Program (2016YFA0100202), the China National Basic Research Program (2011CBB01001,

2011CBA01102, 2009CB941003) and the National Thousand Talents Program of China and the Program for New Century Excellent Talents in University (NCET-11-0482). The funders had no role in study design, data collection and analysis, decision to publish, or preparation of the manuscript.

## Grant Disclosures

The following grant information was disclosed by the authors:
National Key Research and Development Program: 2016YFA0100202.
China National Basic Research Program: 2011CBB01001, 2011CBA01102, 2009CB941003.
National Thousand Talents Program of China.
Program for New Century Excellent Talents in University: NCET-11-0482.

## Competing Interests

The authors declare there are no competing interests.

## Author Contributions

- Yangli Pei conceived and designed the experiments, performed the experiments, analyzed the data, contributed reagents/materials/analysis tools, wrote the paper, prepared figures and/or tables, reviewed drafts of the paper.
- Liang Yue and Wei Zhang performed the experiments, contributed reagents/materials/-analysis tools, reviewed drafts of the paper.
- Jinzhu Xiang performed the experiments.
- Zhu Ma contributed reagents/materials/analysis tools, reviewed drafts of the paper.
- Jianyong Han conceived and designed the experiments, analyzed the data, wrote the paper, prepared figures and/or tables, reviewed drafts of the paper.

## Animal Ethics

The following information was supplied relating to ethical approvals (i.e., approving body and any reference numbers):

All experiments were approved by the Animal Care and the Use Committees of the State Key Laboratories for Agrobiotechnology, College of Biological Sciences, China Agricultural University (Approval number: SKLAB-2016-05-01).

## Data Availability

The raw data is uploaded as Data S1.

## Supplemental Information

Supplemental information for this article can be found online at http://dx.doi.org/10.7717/peerj.4177#supplemental-information.

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
