# Peer review of "Murine pluripotent stem cells that escape differentiation inside teratomas maintain pluripotency"

_PeerJ, doi:10.7717/peerj.4177_

## Round 0.1 · original submission · Major Revisions

· Academic Editor

Major Revisions

The manuscript reports findings that are potentially original and relevant to support the conclusions drawn on the maintenance of pluripotent stem cells in mouse teratoma. However the data shown are technically insufficient and need to be strengthen and further controlled as detailed in the reports from both reviewers.

In particular evidence for Oct4 expression in teratoma needs to be shown via detection of GFP fluorescence and improved staining quality of the histological characterization. In addition, the methodology used lacks significant information and the figures and their legends are incomplete or insufficiently described, outside of the problem of the English language which requires extensive proof reading throughout the manuscript.

Reviewer 1 ·

Basic reporting

The English language needs to be improved throughout all parts of the manuscript to ensure that the text becomes clearly understandable. The language should be reviewed by a native English speaking colleague or a professional service for language editing. I will try to exemplify this for parts of the abstract only: The first half-sentence of the background is understandable; afterwards the reader has to guess what the authors want to say. Methods: abbreviations such as MES, EGFP, iPSCs are not defined and at least MES is an ambiguous abbreviation. Moreover, iPSC are generated by reprogramming, which might involve the transduction of cells by a virus to introduce the reprogramming factors. However, it is wrong to write that iPSCs were generated by virus infection as the authors did. I am also not a native speaker but it is clear that the sentence “And the differentiate ability were study by the assays of in vivo teratoma formation” is understanable or interpretable but it is not a correct English sentence. Results: MES-FT and iPS-FT as MES-ST and iPS-ST are not explained. These abbreviation are even not explained when they are first used in the manuscript, i. e. Fig 2, making the understanding of the manuscript very difficult.

Due the problems regarding the language, it is currently difficult to comment on the scientific background. However, the introductory statement that in vivo differentiation systems are superior to in vitro differentiation systems and even “more adapt to clinical application” is not based on sufficient evidence.

The structure of the article appears to be conform with PeerJ standards. However, the quality of several figures needs to be improved. On all pictures showing cells or tissues scale bars are missing. All figures have very poor legends. Data from which MES and iPS cells are shown in Fig. 2 (is it OCT4-MES and TG iPS1-7)?. In Fig. 2 (TR-PCR) several bands for the housekeeping gene Hprt are not clearly visible making the results for the other genes analyzed questionable. In Figs. 3a, 5c, 5d no specific tissues within the tumors can be identified since the magnification is not sufficient. In Fig 3b, I can identify an endodermal and a mesodermal differentiation but no ectodermal differentiation. Therefore, these pictures do not convincingly demonstrate pluripotency of the cells that were injected to generate these tumors. In Fig. 3c and 5e, the OCT staining is also not very convincing, the brown color looks more like a background staining (and this is different from Figs 3d and 5f). Fig. 7 is, in view of the raw data provided for the qPCR experiment, highly suspicious for showing means plus SD of technical replicates only. Technical replicates would not justify the statistical analysis. Moreover, the use of a Student’s t-test is likely not justified since it requires normal distribution data and equal variations between the data sets. Moreover, which “mES” are shown here? In addition,, some genes are indicated correctly (first letter in capitals) but most others are not. Raw data are supplied for the qPCR data only.

The study by Pei and colleagues is in principle interesting. While many studies have shown the presence of cells expressing pluripotency markers in teratomas (some of which could have been cited), this study suggests that pluripotent cells can be recovered from teratomas and can be sequentially transplanted to generate new teratomas. This has to my knowledge not been shown before (assumed that the tumors are teratomas). However, the tumors shown cannot be clearly identified as teratomas in this manuscript. Diagnosis of a tumor as a teratoma require the presence of tissues derived from all three germ layers and this is not clearly shown. Moreover, the PCR and qPCR data are currently also not convincing (see my comments above).

Experimental design

The design of the study is in principle reasonable and the research question is relevant. The methods used are generally very poorly described (see e.g. “mouse strains” which contains hardly any meaningful information so that strains including the genetic background cannot been identified). Moreover, no antibodies used are identified unambiguously and no information on primers are given. The teratoma experiments are hardly described; it is, e.g., not mentioned to which tissue the cells have been injected or how long the tumors were grown. More similar examples could be given easily.

Validity of the findings

The technical standard of the experiments (or the results presented) is in my view currently not sufficient to justify publication. The authors need to provide data, which unambiguous demonstrate that the tumors are teratomas and they need to provide sound gene expression data.

Currently, the conclusions are rather broad and not very specific with respect to the data shown. However, with respect to the language it is difficult to assess these issues.

Additional comments

The basic critique regarding the experiments (unambiguous demonstration that the tumors are teratomas; providing sound gene expression data) needs to be addressed and the methods needs to be described according to common standards. If this is possible, the authors should work on issues such as language and presentation in the context of the literature.

Reviewer 2 ·

Basic reporting

The English language should be improved. There are many typographical errors, including: s everal, detials, fonud, tarotoma… that could be corrected using Word. Several sentences start with ‘And’; Thus, your manuscript need to be reviewed/edited by a native English speaking colleague.

Another important point is that all acronyms should be clearly defined as they first appear in the text. Some are not defined. For example, what are MES-FT? (I guess that it means Mouse Embryonic Stem- from First Teratoma?).

I suggest that you change the title for: “Pluripotent mouse stem cells persist and can be isolated from in vivo teratoma”.

Background section: To my knowledge, there is no clear demonstration that PSCs derivatives obtained in vivo are superior to derivatives obtained in vitro. So I would remove or at least temper this statement and I would rather write that “Teratoma is a widely used assay in the field of stem cells and regenerative medicine but the cell composition of teratoma is still elusive”.

Experimental design

1. You have used a GFP construct that reports Oct4 expression. First, it seems important to confirm that the GFP signal does report for Oct4 endogenous expression. To tackle this issue, I suggest that you perform immunostaining on your PSC cultures using an anti-OCT4 antibody and a secondary antibody coupled to Cy3 to determine whether or not GFP and Cy3 signals merge. Please note that the correct gene name is Pou5f1, not Oct4, even if Oct4 is widely used and accepted.

2. In your teratoma assays (Figs. 3, 5 and 6), immunostainings were performed using an OCT4 antibody, which could detect endogenous OCT4 that is expressed in adult tissues (see Bhartiya, Stem cells int., 2013). To rule out that you detected Oct4 expressed from the host and to show that your Oct4 signal came from the PSCs that you injected, I suggest that you perform additional stainings on teratoma using an anti-GFP antibody or that you show GFP fluorescence.

3. I suggest that you perform immunostainings on teratoma using antibodies against markers of the three germ layers, as this is the accepted standard in the field.

4. To confirm the pluripotency of your PSC cultures, in particular those issued from teratoma dissection (FT and ST), I suggest that you perform alkaline phosphatase staining.

5. It is not clear whether or not the green signal shown for Oct4 in Figs. 1, 5 and 6 is the fluorescence emitted by GFP driven by Oct4 promoter or whether it comes from an immunostaining? Please explain and mention it in the text.

6. The composition of the different culture media should be detailed in the material and methods part.

7. Scale bars should be included in photographs.

8. Please use an additional housekeeping gene in your PCR experiments (figure 2), as there is a great variability in your Hprt expression (use for example Gapdh, as in your qPCR). Please also provide a legend.

9. I thank you for providing raw qPCR data (Table S1). Your qPCR were performed in triplicates and were correctly analysed. But these triplicates are three measures performed on the same cDNA and therefore it only takes into account the technical variability. To take into account biological variability, RT-qPCR should be performed on at least two more biological samples for each line (so to have n=3 biological samples for each line).You might use Genorm to select three or 4 housekeeping genes to accurately compare expression in your different samples by qPCR (see Vandesompele et al., Genome Biology, 2002, 'Accurate normalization of real-time quantitative RT-PCR data by geometric averaging of multiple internal control genes').

10. It’s not clear if MEFs were removed so that RNA were extracted from pure PSCs samples? Please explain.

Validity of the findings

Overall, the findings are valid and the conclusion are well stated and linked to the original research question.
However, I have found that your conclusion on PGC markers is overstated.Indeed, your PGC markers are weakly expressed in your qPCR assay (levels that of course depend on the efficiency of your primers); Therefore I suggest that you mention that there is an increase in the levels of PGC markers but that they remain weakly expressed. Moreover, it is important to reproduce these findings on more biological samples.

Additional comments

The study by Pei et al. reports the isolation and characterization of pluripotent stem cells (PSCs) from in vivo teratoma. As previously reported for cancer stem cells, the authors found that PSCs can be injected in immunocompromised mice, harvested from teratomas, expanded in vitro, and reinjected several times while keeping some molecular and functional characteristics of PSCs. This is an interesting study that requires a major revision, in particular the English language should be improved and additional experiments must be performed to reach higher standard.

---

## Round 0.2 · Minor Revisions

· Academic Editor

Minor Revisions

The additional experiments and corrections made have strengthened the data and well improved the readability of this revised manuscript.
There remain however, a number of minor points that require the author's attention, as requested by both reviewers. These points concern in particular the addition of missing information on some tools and methods used, a clarification of the statistics relating to data presented in Fig.7 and the addition of a negative control (such as mouse adult fibroblasts) in Figure 2, plus the corrections of remaining spelling mistakes including in the labels for Figure 4A and Figure 6A.

Provided that these points are clarified and corrected, the manuscript should be acceptable for publication.

Reviewer 1 ·

Basic reporting

The authors have made a substantial progress in the presentation of their study. The English is now understandable and the content is mostly clear. The authors should consider further clarification of the sentences in lines 14, 49-50, and 109-110. The figures have been significantly improved. Some abbreviations are not defined at their first appearance (MEF, line 83; DMEM, line 85; FBS, line 85; LIF line 86). The primer sequences should be better provided in a separate table than within the text.

Experimental design

The description of the methods has been improved but some aspects need further attention:
- the secondary antibodies, which have been used, are not identified (lines 98 and 119)
- the description of the qPCR and statistics is mixed up
- the injection site of the stem cells in the NOD/SCID mice is not mentioned

Validity of the findings

The quality of the data has been improved but some aspects need further attention:
- In Fig. 2 a negative control such as MEFs is missing.
- The quality of the histology of the teratomas is much better now. However, the differentiations shown for endoderm in Fig. 3a and ectoderm in Fig. 5c are not completely undisputable in my view. Therefore, I would encourage the authors to do immunohistochemistry for germ layer-specific markers to further support their findings, although this might not be an absolute requirement since the reader can now decide whether she/he wants to follow the interpretation of the authors or not.
- It is essential to reconsider the statistics for Fig. 7: The Wilcoxon test is suitable to compare two paired samples. One might discuss whether samples are paired in these experiments. However, it is clear that always three samples need to be compared. Therefore a Kruskal-Wallis test might be appropriate.

Reviewer 2 ·

Basic reporting

The English language is now correct. But there are still some mistakes that need to be corrected, including:

Takahashiand Yamanaka (a space is missing after the name of the first author)
Teratoma and not Tarotoma as in the panel A of figure 4 and of figure 6.
Nanog and not Naong as written (last page of results)
Mean and not MS in methods
MES-FT and iPS-FT are not defined when they first appear in the text.
SAS program (SAS stands for what?, please define this term)

Experimental design

The anti-GFP antibody that was used is not indicated in the mat and methods


I should have noticed this before but when I read again your manuscript I realize that a negative control line is missing for you PCR and qPCR data.
So please add a negative control, such as fibroblasts, which do not express pluripotency/germ cell markers for PCR in figure 2 and qPCR in figure 7.
.

Validity of the findings

Overall it is good.

However, I would temper and remove ‘highly’ from the sentence “OCT4-positive cells … highly express germ cell markers”. To affirm this, you should have compared the expression levels of these markers in your cells to those in primordial germ cells (positive control) and to a negative control line (such as fibroblasts). So, in your situation, you can conclude that they express germ cell markers, but not that they highly express them.

Additional comments

Your manuscript is now much better. I will add a negative control for your PCR and qPCRs and pay attention to details in your text.

---

## Round 0.3 · Minor Revisions

· Academic Editor

Minor Revisions

The spelling mistakes in the label to Figure 4a and Figure 6a (was Tarotoma instead of Teratoma) is still wrong: it now has been changed to "Terotoma" instead of "Teratoma". Please correct this mistake in Figure 4 and 6. The other responses and corrections made in reply to the reviewer's and editor's comment are acceptable.

---

## Round 0.4 · accepted · Accept

· Academic Editor

Accept

After addition of the last corrections to your manuscript, I am glad to accept it for publication in PeerJ.